# Effectiveness of Multidimensional Controllers Designated to Steering of the Motions of Ship at Low Speed

**DOI:** 10.3390/s20123533

**Published:** 2020-06-22

**Authors:** Witold Gierusz, Monika Rybczak

**Affiliations:** Department of Ship Automation, Gdynia Maritime University, 81-225 Gdynia, Poland; w.gierusz@we.umg.edu.pl

**Keywords:** multivariable system control, linear matrix inequalities, robust control, *H*_∞_ matrix norm and *H*_2_ matrix norm

## Abstract

The article described two full multidimensional controllers applied to steer a real vessel named ‘Blue Lady’ that is used by the Foundation for Safety of Navigation and Environment Protection at its training and research facility loacted at Silm lake in Poland. Both controllers were based on different approaches, but finally gave similar results. The first part describes the object to be controlled which is a training ship used for training of navigators in various conditions, areas and manoeuvres. This is followed by a short description of the theory for both controllers, Robust and Linear Matrix Inequalities (LMI). Next real time trials are described, which are 3 different manouvers for low velocities, executed by both LMI and Robust contrllers. In these trials ‘Blue Lady’ velocities, silhouete trajectory ans wind data are recorded. Finally the quality of work for both controllers is collected in two tables.

## 1. Introduction

Steering the motion of a sea-going ship is one of the most difficult control problems. The main reason is related to the strongly non-linear and non-stationary characteristics of such objects (see e.g., [1]). Additionally, external disturbances play an important role in the whole regulation process since they change reaction of the vessel to steering signals.

There are a few ways to classify control systems used in the area of vessel movement. One of them is the classification in view of the ship’s speed. The discussed control systems can be divided into three types:
1.For exploitation speed i.e., ‘Full ahead’ or a similar one used on open sea:
-stabilization of heading ([2,3,4,5]),-trajectory keeping ([6]),-turning operations ([7]),-roll stabilization ([8,9]),-anti collision maneuvers (although ARPA (Automatic Radar Plotting Aid))systems are move of an advisory than steering type) ([10])
2.For low speed i.e., ‘Slow or Very Slow ahead’ and ‘Slow or Very Slow Astern’ used mainly in constrained areas:
-controlled motion with any drift angle (crab-wise motion) e.g., during berthing, passing narrow channels etc. ([11]),-controlled movement following ROV (Remotely Operated Vehicle) unit ([12]).
3.For speed close to zero:
-dynamic stabilization of position DSP ([12]),-stabilization of ship placement in relation to the hydrodynamic structure, position mooring (e.g., Weather Vaning) ([12]).


From control theory point of view the first class of systems mainly consists of one-dimensional regulators (SISO ones). Although they seem to be rather simple regulators that due to non-linear and non-stationary properties of the ship they often lead to modern and very sophisticated solutions e.g., adaptive or robust controllers.

The second and third class presented above belong to the multi-dimensional types of regulators (the MIMO ones) due to the necessity to steer a few of ship’s velocities simultaneously. Presented paper concentrates on the controllers for small velocities (It should be pointed out that class 2 and class 3 are similar in the sense that they both deal with several velocities. But approaching the controller synthesis in both classes is definitely different. The reason is related to different properties of the ship dynamics when the vessel is at rest or when it moves with non-zero velocity.).

Steering of the ship movement at these velocities in constrained areas is a big challenge for navigational systems due to low accuracy of measuring systems, in turn impacting the precision and safety of ship movement control. Additionally, maneuvering capabilities of ships at low velocities are limited. The power of propulsion systems is relatively small compared to ship mass, which means that a longer time is required to start the movement or change its direction. To achieve precise control of longitudinal, lateral and rotational velocities in such difficult conditions the ship has to be equipped with at least several propulsion systems. At the same time ship’s movement parameters are measured by electro-navigational systems which have strong cross feedbacks. Due to the above constrains it is natural to use multidimensional control systems. It needs to be stressed that multidimentional controlled object identification process is crucial for controller synthesis. In recent years numerous publications have been created that describe this process in marine applications, even for a 4 degrees of freedom (4DOF) object as shown in [13]. Another interesting approach is shown in [14] where identification based on Gauss process is described to be effective. In [15] Support Vector Machines (SVM) optimized by an Artificial Bee Colony (ABC) algorithm are shown. The authors of [16] present numerical simulations that show Jang’s scheme is an effective and very simple method for identification of the characteristics of ship roll motion and produce relatively reasonable solutions. In the 2019 paper [17] an analysis of two methods was performed, a nonlinear Nomoto model and a maneuvering model group (MMG) model of Yupeng ship were established and verified by the turning trial at sea, then an adaptive neuro-fuzzy inference system (ANFIS) controller was trained by learning the actual ship trial data.

Classical multidimensional ship movement control methods include, among others, Robust Controllers using minimization of $H_{\infty }$, described in [18], and controllers using minimization of $H_{2}$ shown in [19], where the authors solve ship autopilot synthesis problems analytically by tuning a corrector to achieve precise ship movement control. With the increase of interest in the above optimal control methods, the interest in numerical methods is also rising, among them Linear Matrix Inequalities that can be used in control system analysis and synthesis. LMI’s are described in detail by authors, for example in [20] and in [21]. In [22,23,24] the authors describe LMI use in multidimensional systems with Fuzzy Logic controllers. In [25] the chapter about applications of a closed loop system shows results of computer simulations for a state space controller controlling the flight path of a missile. In [26] the authors use a Robust LMI controller with $H_{\infty }$ that includes uncertainties, while [27,28,29] describe $H_{2}$/$H_{\infty }$ norm minimization in controller synthesis. Finally [30] show a comparison of μ synthesis and LMI controllers where the controlled object was a ship.

Controller synthesis methods, related to marine industry, presented in majority of publications are purely theoretical or based on computer simulations. The very few publications dealing with actual ships or ship models focus on single dimensional control, trajectory control responsible for moving the ship over a set path. Publications from Rolls Royce [31] and Kongsberg [32] describe much more complex multidimensional control in MASS 4 standard but being published by industrial companies do not contain information about used control methods. In comparison this paper describes multidimensional control of a ship model during actuall sea trails with a detailed description of methods used.

One of the main reasons to adopt the two mentioned approaches (Robust and LMI) to steer the ship was the necessity of fulfilling condition of stability of the whole control system. In multidimensional systems it is not (in general) so easy to prove. In case of Robust controller system stability (and also robust performance) is related to the special form of $H_{\infty }$ norm value called structured uncertainty μ Condition μ<1 guarantees stability [33]. In case of LMI approach one can choose the area (in the left-hand complex plane) to place poles of the closed control system to fulfil the stability condition. Other advantages (taken into account by the authors) were stable and effective algorithms contained in Matlab/Simulink package. They allowed fully multidimensional controllers to simultaneously steer all ship’s thrusters.

## 2. Controlled Object

As a controlled object to verify described regulators one chose an autonomous floating model of a sea-going vessel. It is used for training navigators in different navigational situations e.g., passing narrow channels, berthing, mooring etc.

The vessel named ‘Blue Lady’ is used by the Foundation for Safety of Navigation and Environment Protection at the Silm lake near Ilawa in Poland (see Figure 1). It is an isomorphous model of a VLCC tanker, built from epoxy resin laminate in 1:24 scale. It is equipped with batteryfed electric drives and a two persons control steering post at the stern. Model actuators are driven by DC electric motors and powered by a batteries pack. They are: one main engine, one aft rudder (blade rudder), two tunnel thrusters (fore and aft), two rotating pump thrusters (fore and aft).

The silhouette of the ship is presented in Figure 2.

The main parameters of the ship are as follows:$$\begin{array}{cc} \mathrm{Length}\mathrm{over}\mathrm{all}\;L_{OA} & =\;13.78[m]\\ \mathrm{Beam}\;\;B\;\; & =\;\;2.38[m]\\ \mathrm{Draft}(\mathrm{average})\;\;\text{-}\;\mathrm{load}\mathrm{condition}\;T_{l}\;\; & =\;\;0.86[m]\\ \mathrm{Displacement}\;\;\text{-}\;\mathrm{load}\mathrm{condition}\;\Delta_{l}\;\; & =\;\;22.83[t]\\ \mathrm{Speed}\;V\;\; & =\;\;3.10[kn] \end{array}$$

The high-fidelity, fully coupled, nonlinear simulation model of this ship was built for controllers synthesis using the Matlab package. Special attention was paid to proper modeling of the ship’s behavior during movement with any drift angle (e.g., astern or askew). The full description of the model can be found in [3,34,35,36].

Ship motion parameters and environmental disturbances are measured by navigational equipment installed on board:
-Anschütz Standard 20 gyrocompass,-GILL WindObserver II ultrasonic anemometer,-LEICA DGPS System 500 receiver working in HPN (High Precision Network) mode.

This equipment communicates with the measurement system using serial communication links in NMEA-0183 standard.

During ship dynamics modeling, for controller synthesis, linearization of the model around it’s working point was used. The model identification process took into consideration:
-propulsion operation,-ship hull construction,-stationary Kalman filter system (this system is used for u,v,r velocities recreation) (‘Blue Lady’ ship model was not equipped with equipment for measuring linear velocities thus the need for Kalman filter system),-power distribution system used for calculating three components of vector:
(1)$$u={\left[\tau_{x},\tau_{y},\tau_{r}\right]}^{T}$$
to vector T with seven components of propulsion systems control signals ($ng_{c}$, $\sigma_{c}$, $sstd_{c}$, $sstr_{c}$, $ssod_{c}$, $\alpha_{dc}$, $ssor_{c}$, $\alpha_{rc})$. The details, symbols and ranges of these signals are presented in Table 1. Allocation is not directly the topic of this paper and as such is desribed in detail in [37]

Controlled object has three input signals: $\tau_{x},\tau_{y},\tau_{r}$ and three output signals $\hat{u},\hat{v},\hat{r}$, relation between these signals is shown on Figure 3 Where: [$\tau_{x}$]—required force (thrust) on the ships longitudinal axis, [$\tau_{y}$]—required force (thrust) on the ships lateral axis, [$\tau_{r}$]—required rotational force, [$\hat{u}$]—derivatve longitudinal velocity, [$\hat{v}$]—derivatve lateral velocity and [$\hat{r}$]—derivatve rotational velocity].

During the identification process numerical values of ‘Blue Lady’ model coefficients were calculated, that have been shown in Table 2.

After including mean values of these coefficients the calculated state space model became a nominal (mean) model whose state space and output equations are shown below: $$\left[\begin{array}{c} \dot{x_{1}}\\ \dot{x_{2}}\\ \dot{x_{3}} \end{array}\right]=\left[\begin{array}{ccc} a_{uu} & 0 & 0\\ 0 & a_{vv} & a_{vr}\\ a_{ru} & a_{rv} & a_{rr} \end{array}\right]\times \left[\begin{array}{c} x_{1}\\ x_{2}\\ x_{3} \end{array}\right]+\left[\begin{array}{ccc} b_{uu} & 0 & 0\\ 0 & b_{vv} & b_{vr}\\ b_{ru} & b_{rv} & b_{rr} \end{array}\right]\times \left[\begin{array}{c} \tau_{x}\\ \tau_{y}\\ \tau_{r} \end{array}\right]$$
(2)$$\left[\begin{array}{c} u\\ v\\ r \end{array}\right]=\left[\begin{array}{ccc} 1 & 0 & 0\\ 0 & 1 & 0\\ 0 & 0 & 1 \end{array}\right]\times \left[\begin{array}{c} x_{1}\\ x_{2}\\ x_{3} \end{array}\right]$$

Matrices **A**, **B** and **C** of the controlled object, ‘Blue Lady’ training ship, have the below form: (3)$$A=\left[\begin{array}{ccc} -3.36\times 10^{-5} & 0 & 0\\ 0 & -9.0\times 10^{-3} & -2.0\times 10^{-4}\\ -3.0\times 10^{-3} & -1.0\times 10^{-2} & -7.75\times 10^{-3} \end{array}\right]$$
(4)$$B=\left[\begin{array}{ccc} 3.62\times 10^{-3} & 0 & 0\\ 0 & 2.06\times 10^{-3} & -1.28\times 10^{-5}\\ 3.0\times 10^{-5} & 1.15\times 10^{-5} & 8.0\times 10^{-3} \end{array}\right]$$
(5)$$C=\left[\begin{array}{ccc} 1 & 0 & 0\\ 0 & 1 & 0\\ 0 & 0 & 1 \end{array}\right];D=\left[\begin{array}{ccc} 0 & 0 & 0\\ 0 & 0 & 0\\ 0 & 0 & 0 \end{array}\right]$$

The ship is equipped with several thrusters and a blade rudder (see Figure 1). Hence the 8 input signals can be used in a control system synthesis. The details of these signals are presented in Table 2.

Presented model of ship dynamics was supplemented by the models of dynamics of thrusters and blade rudder, Kalman filters for reconstruction of ship velocities and models of forces driving from wind.

## 3. Robust Controller

### 3.1. The Augmented Plant

One of the latest approaches is based on minimization of matrix norm $H_{\infty }$ (see e.g., [33]) and is shortly described below.

The feedback controller design can be formulated using the concept of uncertainties and weighting functions. It is a convenient way of introducing unmodelled object dynamics and different signal specifications into a MIMO process:
-such quantities can be used for different purposes e.g., establishing the requirements for control quality, description of disturbances, introduction of the influence of non-modelling dynamics of the plant etc.,-the signals scaling operation is easy to perform by means of weighting functions,-one can distinguish between more and less important components of the signals vectors (e.g., in errors vector) by proper gain coefficients,-designer requirements related to the particular signals can be formulated for specified frequency ranges in a natural way.

In the classic control theory the assumed steering quality is only achieved when the plant model is accurate. If the real plant differs (e.g., due to operating conditions) from the model used during controller synthesis this quality can be significantly poorer. The differences between the object and the model are usually named system uncertainties. There are several sources of uncertainties: changes to the physical parameters of the vessel due to different work conditions (e.g., load, trim, depth of water, etc.), neglected nonlinearities inside the object (e.g., related to hydrodynamics phenomena), unmodelled dynamics, especially in high frequency range etc. The uncertainties can be represented by means of two components:
-the first one is the “pure” uncertainty Δ, bounded in the $H_{\infty }$ norm sense, i.e., ${∥\Delta ∥}_{\infty }\leq 1$-the second one it is the weighting function modeling the magnitude and shape of the uncertainty in the frequency domain.

The whole model of the ship’s dynamics with uncertainties and weighting functions is called the augmented model of the plant. Then the closed-loop control diagram can be presented as follows (see Figure 4 and note opposite directions of signals—from right to left hand side, more convenient for matrix operations used in multivariable systems).

Different weighting functions have different meaning. Functions matrices $W_{s}$ and $W_{u}$ define designer requirements for steering quality in the system while functions matrices $W_{p}$, $W_{n}$ and $W_{z}$ form input signals in the frequency domain (weighting functions related to the uncertainties are included ’inside’ the plant matrix G). One can write the following equations based on Figure 4:(6)$$\begin{array}{ccc} e_{y} & = & W_{s}W_{z}\tilde{r}-W_{s}W_{p}\tilde{p}-W_{s}\mathrm{Gu} \end{array}$$(7)$$\begin{array}{ccc} e_{u} & = & W_{u}u \end{array}$$(8)$$\begin{array}{ccc} v & = & W_{z}\tilde{r}-W_{p}\tilde{p}-W_{n}\tilde{n}-W_{s}\mathrm{Gu} \end{array}$$(9)$$\begin{array}{ccc} u & = & Kv \end{array}$$

Above equations can be rewritten in a more compact form: (10)$$\left[\begin{array}{c} e_{y}\\ e_{u}\\ v \end{array}\right]=P\times \left[\begin{array}{c} \tilde{r}\\ \tilde{p}\\ \tilde{n}\\ u \end{array}\right]$$
where matrix **P** has the form:(11)$$P=\left[\begin{array}{cccc} W_{s}W_{z} & -W_{s}W_{p} & 0 & -W_{s}\mathrm{Gu}\\ 0 & 0 & 0 & W_{u}\\ W_{z} & -W_{p} & -W_{n} & -W_{s}\mathrm{Gu} \end{array}\right]$$

Matrix P is called the augmented plant (model plant) due to all weighting functions included in it. Introducing the input vector $d={\left[\;\tilde{r}\;\;\tilde{p}\;\;\tilde{n}\;\right]}^{T}$ and the weighting error vector $e={\left[\;e_{y}\;\;e_{u}\;\right]}^{T}$ one can write:(12)$$\begin{array}{ccc} \left[\begin{array}{c} e\\ v \end{array}\right] & = & P\times \left[\begin{array}{c} d\\ u \end{array}\right] \end{array}$$(13)$$\begin{array}{ccc} u & = & K\times v \end{array}$$

The last two equations with the matrix of the pure uncertainties Δ enable to build the generalized configuration of the control system shown in Figure 5.

### 3.2. The Controller Synthesis

The augmented plant **P** consists of the nominal object model $G_{n}$ and of all matrices of weighting functions (modeling the performance requirements, forming input signals and describing the uncertainties).

The weighting error vector can be expressed in the form:(14)$$e=T_{\mathrm{ed}}\left(P,K\right)\cdot d$$
where matrix $T_{\mathrm{ed}}$ can be obtained by means of the Lower Linear Fractional Transformation (see [38]).

The control system synthesis can be treated as a process of calculating a controller K which maintains small certain weighted signals (e.g., control errors) under assumption that the changes of the plant (ship) properties are inside the determined area (described by weighting functions of uncertainties). One of the possible ways to define the ’smallness’ of signals (or transfer matrices) are matrix norms $H_{\infty }$ (see e.g., [39]) expressed by the following equations:(15)$$∥T_{\mathrm{ed}}\left(s\right){∥}_{\infty }=\max_{\omega \in \langle 0,\infty \rangle }\bar{\sigma }\left[T_{\mathrm{ed}}\left(j!\right)\right]$$

The algorithm named ‘D-K iteration’ from Matlab was used to compute the robust $H_{\infty }$ controller for the system presented in Figure 4. The obtained controller in state model form was of high order equal to 41—the same as the open-loop system. The stablility of the whole closed-loop control system is ensured by execution of the modified Small Gain Theorem. For described system (given by matrix $T_{\mathrm{ed}}$ ) with uncertainties one can couch so called Structured Singular Value—$T_{\mathrm{ed}}$ (see Equation (Equation 15)). Without a loss in generality, when $\mu_{T_{\mathrm{ed}}}$ is smaller the 1, the control system is internally stable (more details one can found in Skogestad and Postlethwaite (2003) with the proofs of the proper theorems).

The value of $H_{\infty }$ norm was 0.56<1 which ensures the robust property of the controller.

The order reduction procedures were performed to decrease this order (without the decline of robust property). Finally a controller of the 21st order was obtained.

The description of the whole procedure named ‘D-K iteration’ and ‘Optimal Hankel Norm Approximation’ essential to obtain the Robust controller is too extensive for presented article, therefore only the set of conditions was presented. The procedures were described in details in the paper: [40].

## 4. LMI Controller

### 4.1. The LMI Concept and the Formulation of the Plant

Linear Matrix Inequalities are a convex optimization tool used, among others, for controller synthesis based on robust control ([20]). Based on LMI’s canonical form:(16)$$\;F\left(x\right)=F_{0}+\sum_{i=1}^{m}F_{i}\cdot x_{i}≻0\;$$
where:
-decision variable vector (unknown) *x* = ${\left[x_{1},x_{2},\ldots ,x_{m}\right]}^{T}$$\in R^{m}$,-matrices marked as $F_{0}\ldots Fi\in R^{nxn}$ are real and symmetrical,-the term “≻0” means that the matrix F(x) is positively defined.

Lyapunov noticed that the necessary and sufficient condition for a linear system to be asymptotically stable is finding a symmetrical positively defined matrix $P=P^{T}$, P≻0 (The problem of finding a symmetrical, positively defined matrix **P** is often called the feasibility problem.) (**P** is the unknown variable) while fulfilling the Lyapunov inequality:(17)$$A^{T}P+\mathrm{PA}≺0$$

Combining the above condition about the feasibility problem with P≻0 relation the LMI stability condition can be formed:(18)$$\left[\begin{array}{cc} -A^{T}P-\mathrm{PA} & 0\\ 0 & P \end{array}\right]≻0$$
where matrix **A** is the controlled object and matrix **P** is the unknown.

This problem is described in [20,21,25,41]. The ‘Blue Lady’ multidimensional model is described in detail also in e.g., [42]. The controller synthesis method for a state space controler for above ship model is shown in [36,43]. Based on simulation results controller structure based on a state space controller was chosen, as:(19)$$u=K\cdot x_{zk},$$

Closed loop system state space equations have the form of (due to the complexity of the problem, in this part the sizes of specific matrices have not been given. However state space variable $x_{zk}$ is composed of state space variables of the inertial element and controlled object):(20)$$\left\{\begin{array}{cccccccc} {\dot{x}}_{zk} & = & \left(A_{cl}+B_{u}K\right)x_{zk} & + & B_{w}w, & x_{zk} & = & \left[\begin{array}{c} x_{i}\\ x \end{array}\right],\\ z & = & \left(C_{z}+D_{zu}K\right)x_{zk} & + & D_{zw}w\\ z_{\infty } & = & \left(C_{\infty }+D_{\infty u}K\right)x_{zk} & + & D_{\infty z}w\\ z_{2} & = & \left(C_{2}+D_{2u}K\right)x_{zk} & + & D_{2w}w, \end{array}\right.$$
where:
-$A_{cl}$—controlled object state space matrix, represents system dynamics;-$B_{u}$—control matrix of control ‘*u*’ signal ;-$B_{w}w$—control matrix of input ‘*w*’ signal ;-$C_{z}$—output matrix of output signal ‘*z*’;-$C_{\infty }$—output matrix of ‘$z_{\infty }$’ signal;-$C_{2}$—output matrix of ‘$z_{2}$’ signal;-$D_{zu}$—transition matrix of ‘*z*’ and ‘*u*’ signals;-$D_{zw}$—transition matrix of ‘*z*’ and ‘*w*’ signals;-$D_{\infty u}$—transition matrix of ‘$z_{\infty }$’ and ‘*u*’ signals;-$D_{\infty w}$—transition matrix of ‘$z_{\infty }$’ and ‘*w*’ signals;-$D_{2u}$—transition matrix of ‘$z_{2}$’ and ‘*u*’ signals;-$D_{2w}$—transition matrix of ‘$z_{2}$’ and ‘*w*’ signals;-‘$z_{\infty }$’ and ‘$z_{2}$’—additional output signals necessary for calculations of $H_{\infty }$ and $H_{2}$ norms.

Where matrices $A_{cl}$, $B_{w}w$ and $B_{u}$ have the form of:$$A_{cl}=\left[\begin{array}{cc} A_{in}-B_{in}{\mathrm{DC}}_{in} & -B_{in}C\\ B_{np1}C_{in} & A_{np1} \end{array}\right],B_{w}w=\left[\begin{array}{c} B_{in}\\ 0_{\left[3x3\right]} \end{array}\right],B_{u}=\left[\begin{array}{c} B_{in}D\\ B_{np1} \end{array}\right].$$

In the following sections of this paper the authors will often relate to state space equations of the closed loop system (Equation 20) in which matrices have been defined taking into account parametric uncertainties $A_{np1}$, $B_{np1}$ with a first order inertial element included in the system (described with matrices $A_{in}$, $B_{in}$, $C_{in}$ and $D_{in}$). Figure 6 shows a simplified block diagram of the control system with an LMI state space controller.

Structure of the closed loop system from Figure 6 includes input signal vector *w*, control signal vector *u* and output signal vectors $z,z_{2},z_{\infty }$.

However, $z_{\infty }$ and $z_{2}$ signal vectors are symbolic and have been implemented in state space equations in order to determine $H_{\infty }$ and $H_{2}$. The methodology of the controller synthesis based on Linear Matrix Inequality consists of a few steps which have to be fulfilled simultaneously.

### 4.2. Stability Restriction—Poles in the Left Half-Plane of Complex Variable Plane s

The first stage is to define the pole placement region located in the left half-plane of complex variable plane s in order to specify dynamic parameters of the designed closed loop system. First LMI condition:$$R_{D}\left(A,X_{D}\right)=L⊗X_{D}+M⊗\left(AX_{D}\right)+$$
(21)$$+M^{T}⊗{\left(AX_{D}\right)}^{T}≺0$$
is fulfilled if and only if there exists a symmetrical positively defined matrix $X_{D}$. where:
-A—in this specific case instead of matrix A we have to use the form $=(A_{cl}+B_{u}K)$, has to be used,-$X_{D}$—the unknown, symmetrical positively defined Lyapunov matrix $X≻0,X=X^{T}$,-L,M—specific, user defined matrices.

The choice of the shape of the convex region and its parameters (e.g., circle with diameter r and center q or location of vertical stripes $a_{1}$ and $a_{2}$) was done empirically by the user. Influence of pole placement on controller parameters is shown in [25,44,45].

### 4.3. Minimization of $H_{\infty }$ Matrix Norm

The second stage is $H_{\infty }$ minimization related to an estimation of scalar value $\gamma_{\infty }$ which is the upper limit of the $H_{\infty }$ norm. When determining the smallest value of $\gamma_{\infty }$ for the given pole placement region of the closed loop system, it was assumed that the obtained scalar value $\gamma_{\infty }$ is constant. Scalar value $\gamma_{\infty }$ can be calculated based on $H_{\infty }$ minimization or it can be assumed to be a constant limit. Second LMI condition has the form:(22)$$\left[\begin{array}{ccc} AX_{\infty }+X_{\infty }A^{T} & B & {\mathrm{XC}}^{T}\\ B^{T} & -\gamma_{\infty }^{2}I & D^{T}\\ \mathrm{CX} & D & -I \end{array}\right]≺0$$

The above equation is generic and in this specific case, taking into account Equation (Equation 20), it has the below form:
-A = $\left(A_{cl}+B_{u}K\right)$;-B = $B_{w}$;-C = $C_{\infty }+D_{u\infty }K_{\infty }$;-D = $D_{\infty w}$.

### 4.4. Minimization of $H_{2}$ Matrix Norm

The third stage is $H_{2}$ norm minimization. One of the ways of obtaining this is to look for $\gamma_{2}$. Assuming that:(23)$$\gamma_{\infty }>\gamma_{\infty min}$$

Value $\gamma_{2}$ can be obtained using the Pareto Curve (see [29,46,47]). We receive a relation between the set value of $H_{\infty }$ norm (and more precisely the minimal value of $\gamma_{\infty }$) and a minimized value of $H_{2}$ norm (which is the value of $\gamma_{2}$). The third LMI condition:(24)$$\left\{\begin{array}{c} \left[\begin{array}{cc} AX_{2}+X_{2}A^{T} & B\\ B^{T} & -I \end{array}\right]≺0\\ \left[\begin{array}{cc} Q & C^{T}\\ C & X_{2} \end{array}\right]≻0\\ Tr\left(Q\right)<\gamma_{2}^{2}, \end{array}\right.$$
is fulfilled if and only if there exists a symmetrical positively defined matrix $X_{2}$ and a symmetrical matrix Q. The above equation is generic and in this specific case, taking into account Equation (Equation 20), it has the below form:
-A = $\left(A_{cl}+B_{u}K\right)$;-B = $B_{w}$;-C = $C_{2}+D_{2u}K_{2}$;-D = $D_{2u}$.

### 4.5. Last Assumption and Final Controller

The fourth stage uses Theorem 10.1 (Equation 25) from [25] where the authors assume that matrix X is equal to $X=X_{D}=X_{2}=X_{\infty }$. In Duan and Yu (2013) there is a following proposition:

The problem of finding a controller is possible if and only if the below set:$$P=\left\{\left(X_{D},Y_{D},X_{2},Y_{2},X_{\infty },Y_{\infty }\right)\in \left\{\begin{array}{c} \left(X_{D},Y_{D}\right)\in \Omega_{D}\\ \left(X_{2},Y_{2}\right)\in \Omega_{2}\\ \left(X_{\infty },Y_{\infty }\right)\in \Omega_{\infty } \end{array}\right\};\ldots \right.$$
(25)$$\left.\ldots Y_{D}X_{D}^{-1}=Y_{2}X_{2}^{-1}=Y_{\infty }X_{\infty }^{-1}\right\}$$
is not null, and in this case, a gain matrix is determined by:(26)$$K=Y_{\infty }X_{\infty }^{-1}=Y_{2}X_{2}^{-1}=Y_{D}X_{D}^{-1}.$$

For parameter set P finding the matrix:(27)$$(X_{D},Y_{D},X_{2},Y_{2},X_{\infty },Y_{\infty })\in P,$$
is not an LMI problem and therefore might be very difficult to perform. Thus the matrix is defined assuming that: (28)$$X_{D}=X_{2}=X_{\infty }≜X,$$
and
(29)$$Y_{D}=Y_{2}=Y_{\infty }≜Y,$$
than parameter set P is reduced to:$$P_{0}=\left\{\left(X,Y,X,Y,X,Y\right);\left(X,Y\right)\in \Omega_{D},\left(X,Y\right)\in \Omega_{2},\left(X,Y\right)\in \Omega_{\infty }\right\}$$
$$=\left\{\left(X,Y,X,Y,X,Y\right)\;;\;\left(X,Y\right)\in \Omega_{D}\cap \Omega_{2}\cap \Omega_{\infty }\right\}$$
(30)$$=\left\{\left(X,Y,X,Y,X,Y\right)\;;\;\left(X,Y\right)\in \Omega_{0}\right\}.$$

Based on the above relation matrices $\left(X,Y\right)\in \Omega_{0}$ can be found rather than $(X_{D},Y_{D},X_{2},Y_{2},X_{\infty },Y_{\infty })\in P$ and gain matrix has the form of:(31)$$K={\mathrm{YX}}^{-1}.$$

This means that LMI conditions formulated for $H_{\infty }$ by Equation (Equation 22), for $H_{2}$ by Equation (Equation 24) and for chosen pole placement (Equation (Equation 21)) are fulfilled if there exists a symmetrical positively determined Lyapunov Matrix. This assumption (extortion), that all Lyapunov matrices are equal in all variants and that all three LMI conditions are treated as one limitation, allows to perform stat space controller synthesis. This “method” leads to the so called “conservative” optimization, which means that some of the conditions will be fulfilled with a large margin (often called a conservative solution). Because matrix **X** is positively determined there exists its converse matrix **Y** and so state space controller matrix K can be described as:(32)$$K=YX^{-1}\cdot$$

Taking into account proposition (Equation 25) a single general LMI condition was formulated (the described area of limitation is marked $\Omega_{0}$ on Figure 7) thanks to which the above method allowed to calculate state space controller gain matrix $K=YX^{-1}$.

The final form of state space controller gain matrix K, which stabilizes the whole control system and minimizes $H_{\infty }$ and $H_{2}$ norms, is shown below:(33)$$K=\left[\begin{array}{cccccc} 1498.85 & 0.05 & 0.27 & -799.67 & -0.02 & -0.16\\ -0.03 & 1696.22 & -5.115 & 0.02 & -952.02 & 3.26\\ -5.68 & -1.63 & 440.99 & 3.44 & 1.03 & -247.15 \end{array}\right]$$

The received matrix is a suboptimal solution also know as the conservative solution. It comes from the assumption that $X=X_{D}=X_{2}=X_{\infty }$ and is the sufficient solution.

## 5. Experiments

### 5.1. The Steering Quality Validation

Presented regulators are compared during real - time experiments performed on the lake Silm near Ilawa Poland. The training ship ‘Blue Lady’ was used as a control object.

A few exemplary results are presented below. The quality of the steering was calculated using two values: mean error of the three velocities (see Equation (Equation 34)) and the maximum range of the deviation calculated for the controlled signal in the steady state period of the simulation runs (see Equation (Equation 35)):(34)$$\Delta x\left[\%\right]=\frac{\sqrt{\sum_{n}^{1}{\left(x_{given}-x_{control}\right)}^{2}}}{n}\cdot 100,$$
where: *n*—number of measurements; *x*—deviation value, in percentage, of measured parameter x∈u,v,r; $x_{given}$—reference signal value (given value) $x_{control}$—output signal value received from the system.
(35)$$\Delta_{u,v,r}\left[\%\right]=\left(x_{max}-x_{min}\right)\cdot 100,$$
where:
-$x_{max}$—maximum output signal value received from the system,-$x_{min}$—minimum output signal value received from the system.

In each of the presented results, called manouvers 1, 2 and 3 respectively, the first figure shows ‘Blue Lady’ ship model silhouete trajectory in *x* and *y* coordinate system as measured by the GPS.

The second and third figures show results for both controllers and for comparison reasons both have been done in the same time frame.

All [*u, v, r*] velocity graphs also include wind reading, (taken from GILL Wind Observer II) shown in Beaufort scale.

### 5.2. Exemplary Results of Three Exercises

-Maneuver no. 1Ahead movement with a given longitudinal velocity *u* = 0.1 [m/s], with maneuver of the duration of 700 [s]. Values of lateral and rotational velocities were set to *v* = 0 [m/s], *r* = 0 [deq/s]. This type of ship maneuver, at low velocities, is very common when navigating narrow passages like channels, rivers or entering harbors. Trajectory ship are shown on Figure 8. Results of the Robust controller are shown on Figure 9, and for the LMI controller on Figure 10.-Maneuver no. 2Ahead and sideways movement with given longitudinal velocity *u* = 0.1 [m/s] and lateral one *v* = 0.05 [m/s] with no rotational velocity *r* = 0 [rad/s] and with maneuver duration of 500 [s]. This type of ship maneuver, at low velocities, is a typical approach to berth which is located parallel to ships. Trajectory ship are shown on Figure 11. Results of the Robust controller are shown on Figure 12 position, and for the LMI controller respectively on Figure 13.-Maneuver no. 3Ahead and sideways movement with rotation when performance velocities were set as follows. Duration of the maneuver was 1100 [s]. This type of ship maneuver, at low velocities, is a typical approach to berth which is located perpendicular to ships course. Trajectory ship are shown on Figure 14. Results of the Robust controller are shown on Figure 15, and for the LMI controller on Figure 16.

### 5.3. Comparison and Comments

Based on presented Figure 8, Figure 9, Figure 10, Figure 11, Figure 12, Figure 13, Figure 14, Figure 15 and Figure 16 one can calculate the quality of the work of both controllers. Taking into account formulas (Equation 34) and (Equation 35) one can create tables (Table 3 and Table 4 respectively) where the numerical comparison of Robust controller and LMI one is presented. The main conclusion is as follows: both controllers have similar quality of work but the LMI controller is simpler.

The controller matrix size for the Robust controller is [21x14] while for LMI controller it has a dimension of only [3x6]. It proves that using LMI approach according to the assumptions of [25] can reduce the controller size. Simulation results show also that LMI controllers are simple and effective control systems.

It was observed that atmospheric factors, like wind, didn’t have a meaningful effect on controller operation. Unfortunately GPS operation mode fluctuations had a severe impact on controller operation. In several areas of Silm Lake the GPS operation mode changed itself which resulted in “repositioning” the ship by several meters in calculations. This effect can be observed on maneuver no. 2 for the Robust controller between 50 [s] and 100 [s] as well as in maneuver no. 3 for Robust controller in 700 [s]. Multidimensional controllers can also be used in small ships used in autonomous transport systems in coastal and inshore waters [48,49,50].

## 6. Conclusions

This paper presents synthesis methods for a multidimensional Robust controller and a multidimensional LMI controller. Many real experiments have been performed on the lake and examples useful for examining both controllers performance have been presented.

It should be pointed out that both controllers ensure stability of the whole control systems, what it not so obvious (and not easy to prove) in case of multidimensional controllers.

The testing will be done while navigating narrow passages and berthing both parallel and perpendicular to ships heading. It is very instructive to compare the quality of the same manoeuvres performed by experienced ship masters and computers.

## Figures and Tables

**Figure 1 sensors-20-03533-f001:**
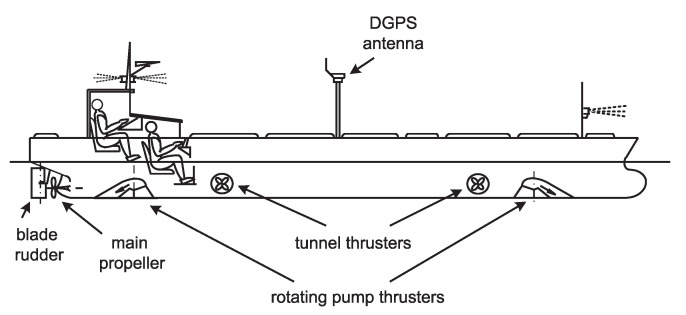
The outline of the training ship ‘Blue Lady’.

**Figure 2 sensors-20-03533-f002:**
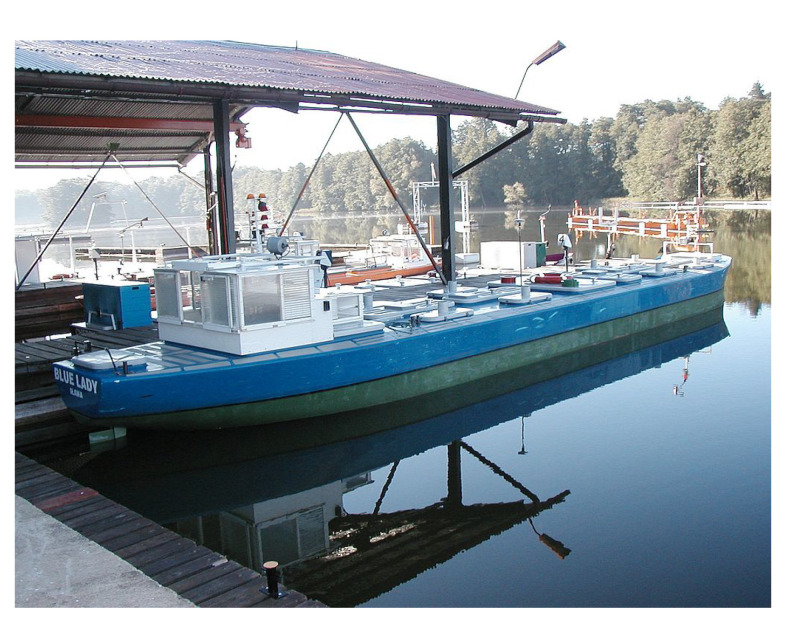
Picture of lardboard the training ship ‘Blue Lady’.

**Figure 3 sensors-20-03533-f003:**
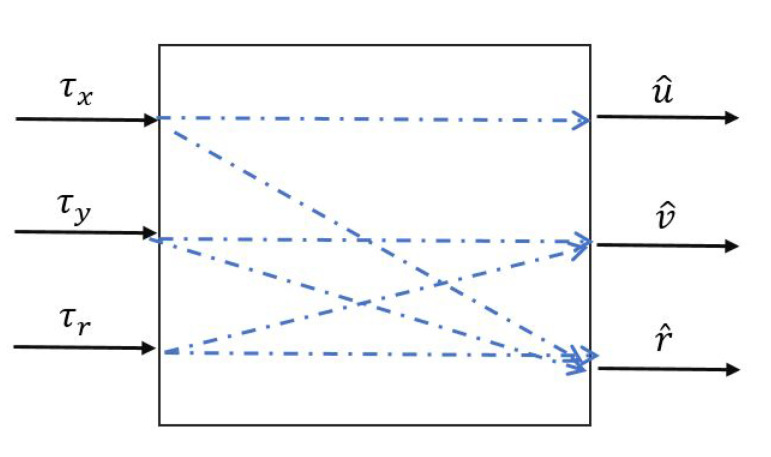
Corelaction with signals.

**Figure 4 sensors-20-03533-f004:**
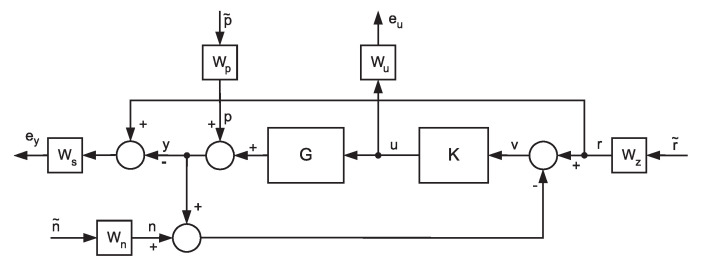
The block diagram of the closed-loop system with the plant with uncertainties and weighting functions for selected signals. The meaning of the particular signals is as follows: $\tilde{r}$ = references vector, $\tilde{p}$—vector of disturbances, $\tilde{n}$—noises vector, ${\tilde{e}}_{y}$—weighted control errors, ${\tilde{e}}_{u}$—weighted control signals.

**Figure 5 sensors-20-03533-f005:**
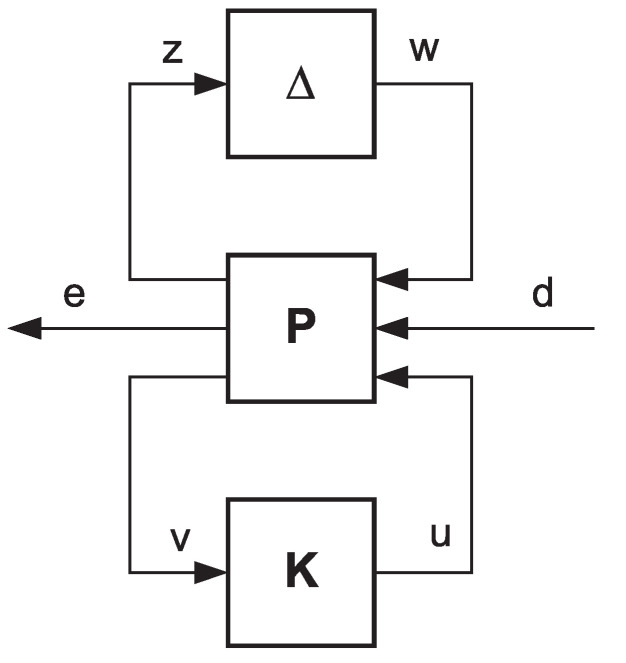
The generalized closed-loop system configuration with uncertainties.

**Figure 6 sensors-20-03533-f006:**
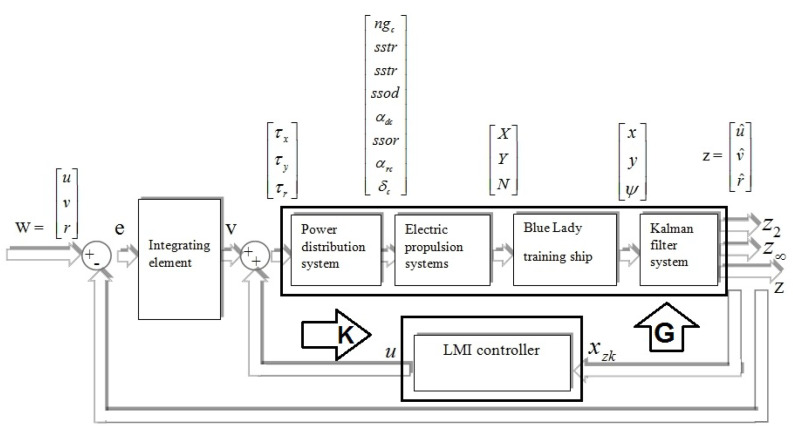
Simplified block diagram of the closed loop control system.

**Figure 7 sensors-20-03533-f007:**
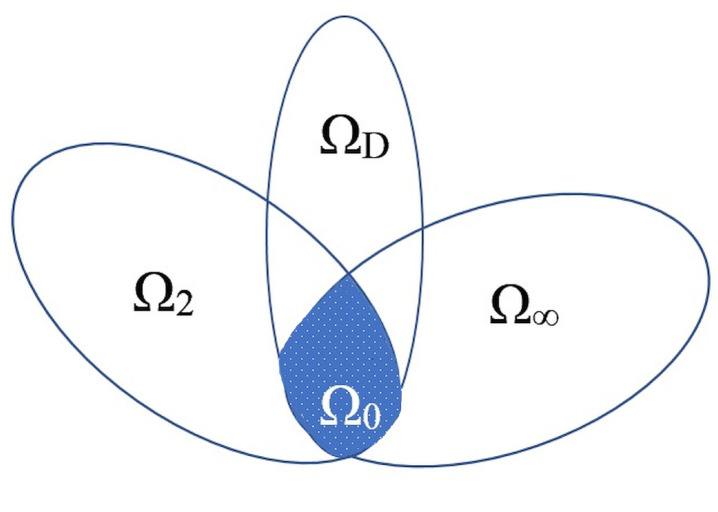
Parameter set $\Omega_{2},\Omega_{\infty },\Omega_{D}$, areas marker is $\Omega_{0}$.

**Figure 8 sensors-20-03533-f008:**
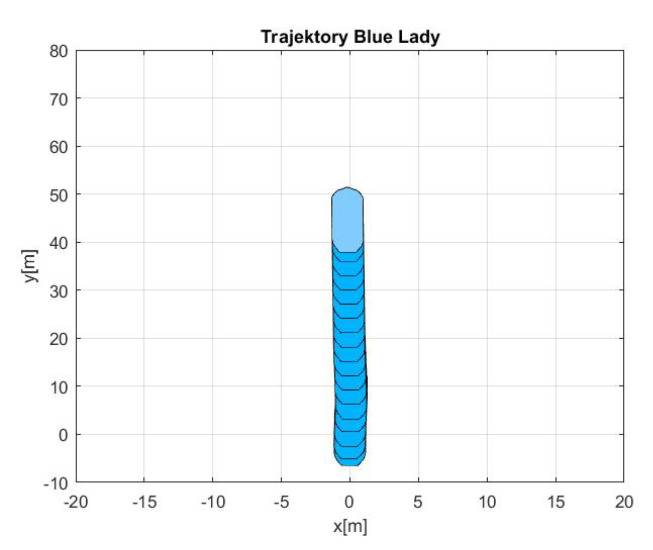
Trajectory ship ‘Blue Lady’ for ahead maneuver with longitudinal velocity *u* = 0.1 [m/s], *v* = 0.0 [m/s], *r* = 0.0 [rad/s] and maneuver duration 500 [s].

**Figure 9 sensors-20-03533-f009:**
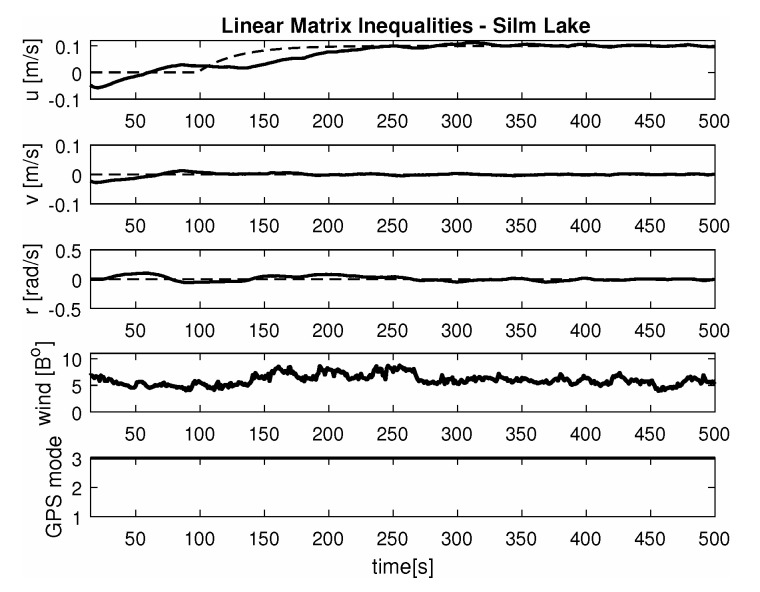
Ahead maneuver with longitudinal velocity *u* = 0.1 [m/s], *v* = 0 [m/s], *r* = 0 [rad/s], wind $\left[B^{o}\right]$ and GPS mode [-], maneuver duration 500 [s].

**Figure 10 sensors-20-03533-f010:**
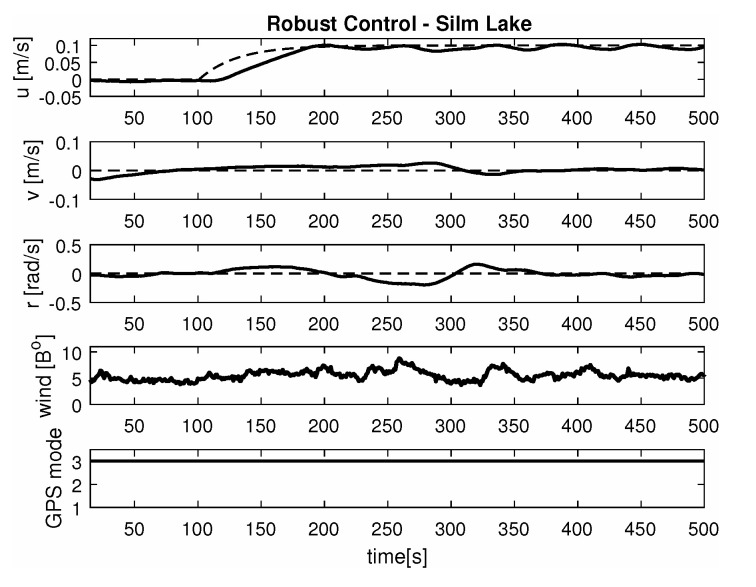
Ahead maneuver with longitudinal velocity *u* = 0.1 [m/s], *v* = 0 [m/s], *r* = 0 [rad/s], wind $\left[B^{o}\right]$ and GPS mode [-], maneuver duration 500 [s].

**Figure 11 sensors-20-03533-f011:**
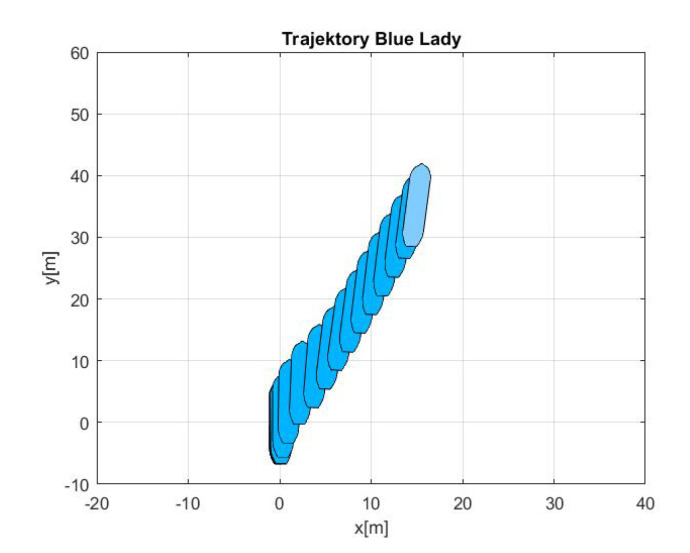
Trajectory ship ‘Blue Lady’ for ahead maneuver with longitudinal velocity *u* = 0.1 [m/s], *v* = 0.05 [m/s], *r* = 0.0 [rad/s], wind $\left[B^{o}\right]$ and GPS mode [-], maneuver duration 500 [s].

**Figure 12 sensors-20-03533-f012:**
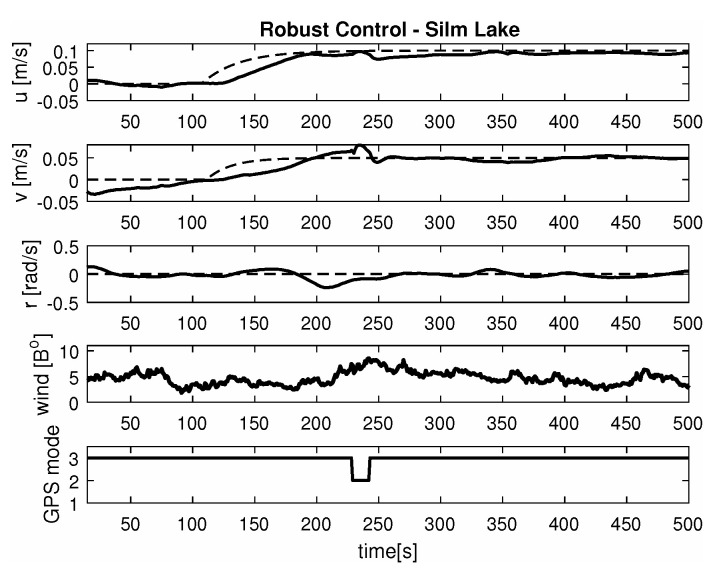
Ahead maneuver with longitudinal velocity *u* = 0.1 [m/s], *v* = 0.05 [m/s], *r* = 0.0 [rad/s], wind $\left[B^{o}\right]$ and GPS mode [-], maneuver duration 500 [s].

**Figure 13 sensors-20-03533-f013:**
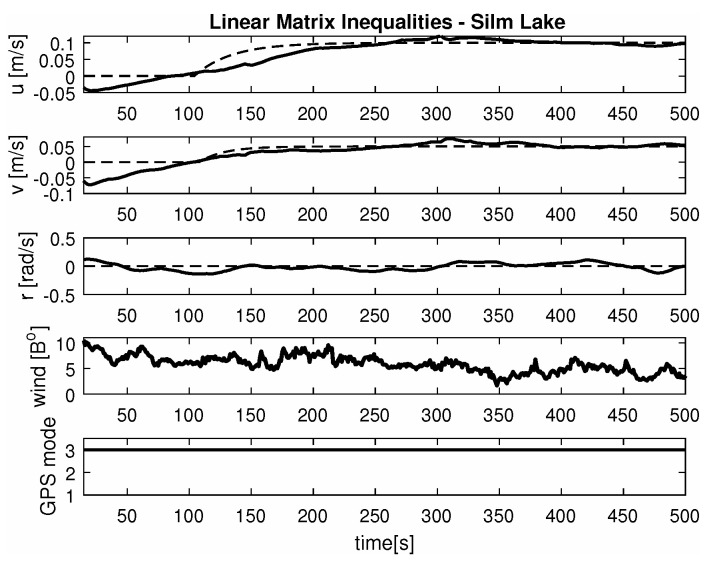
Ahead maneuver with longitudinal velocity *u* = 0.1 [m/s], *v* = 0.05 [m/s], *r* = 0.0 [rad/s], wind $\left[B^{o}\right]$ and GPS mode [-], maneuver duration 1100 [s].

**Figure 14 sensors-20-03533-f014:**
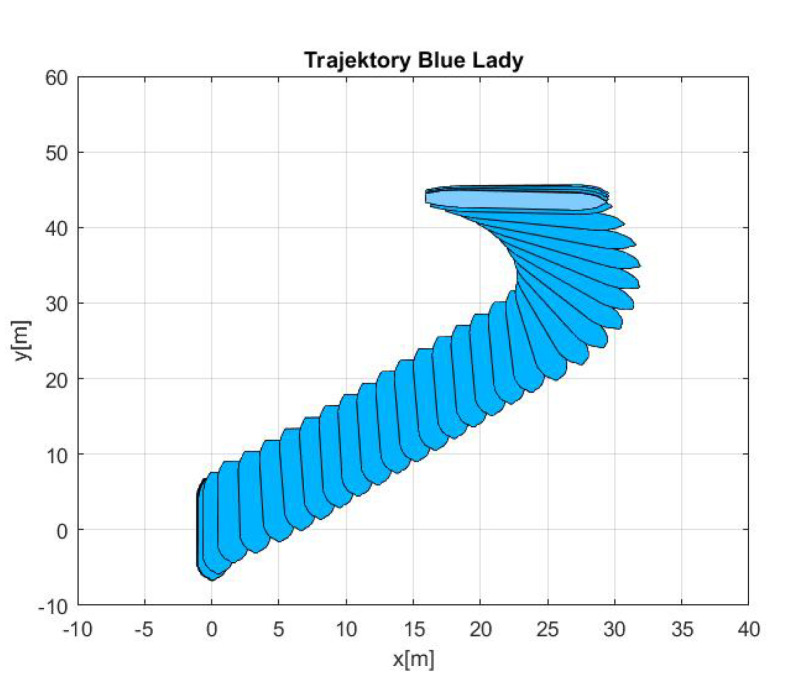
Trajectory ship ‘Blue Lady’ for ahead maneuver with longitudinal velocity *u* = 0.05 [m/s], *v* = 0.05 [m/s], *r* = −0.3 [rad/s] wind $\left[B^{o}\right]$ and GPS mode [-], maneuver duration 1100 [s].

**Figure 15 sensors-20-03533-f015:**
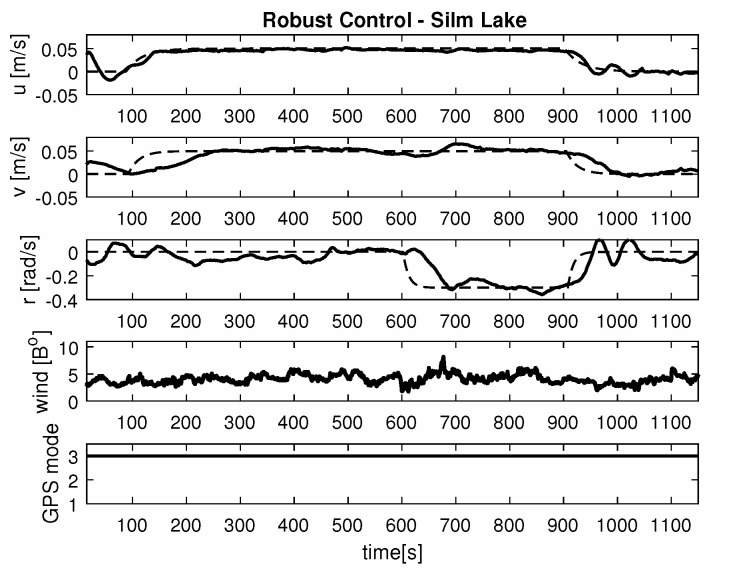
Ahead maneuver with longitudinal velocity *u* = 0.1 [m/s], *v* = 0.05 [m/s], *r* = −0.3 [rad/s], wind $\left[B^{o}\right]$ and GPS mode [-], maneuver duration 1100 [s].

**Figure 16 sensors-20-03533-f016:**
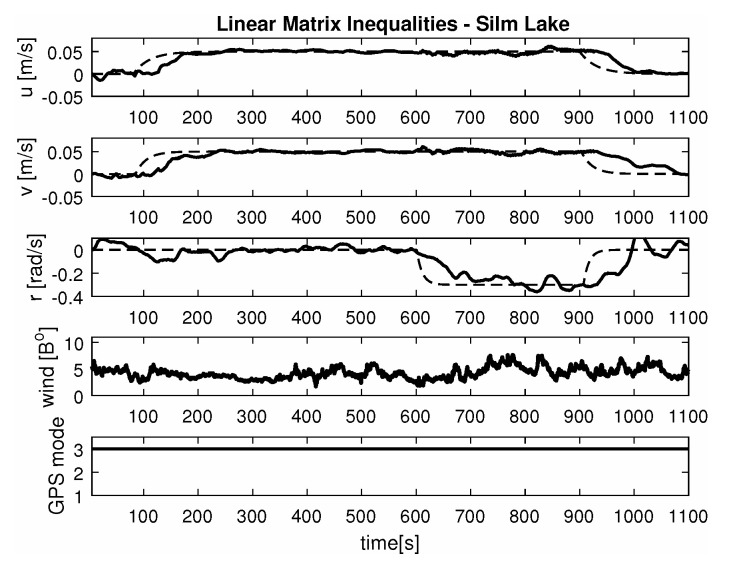
Ahead maneuver with longitudinal velocity *u* = 0.05 [m/s], *v* = 0.05 [m/s], *r* = −0.3 [rad/s], wind $\left[B^{o}\right]$ and GPS mode [-], maneuver duration 1100 [s].

**Table 1 sensors-20-03533-t001:** Characteristics of the input signals for ship’s propellers and rudder [11].

No	Signal	Symbol	Range	Unit
1	revolutions of the main propeller	*ng_c_*	[−200–480]	[rpm]
2	conventional rudder angle	$\delta_{c}$	[−35–35]	[deg]
3	relative thrust of the bow tunnel thr.	*sstd_c_*	[−1–1]	[-]
4	relative thrust of the stern tunnel thr.	*sstr_c_*	[−1–1]	[-]
5	relative thrust of the bow pump thr.	*ssod_c_*	[0–1]	[-]
6	turn angle of the bow pump thr.	$\alpha_{dc}$	[−120–120]	[deg]
7	relative thrust of the stern pump thr.	*ssor_c_*	[0–1]	[-]
8	turn angle of the stern pump thr	$\alpha_{rc}$	[60–300]	[deg]

Symbols shown in the third column are used throughout the whole paper.

**Table 2 sensors-20-03533-t002:** Ship model coefficients values.

Coeff.	Value Min.	Value Max.	Mean Value
$a_{uu}$	$-5.99\times 10^{-3}$	$-0.72\times 10^{-3}$	$-3.36\times 10^{-3}$
$a_{vv}$	$-1.40\times 10^{-2}$	$-0.40\times 10^{-2}$	$-9.00\times 10^{-3}$
$a_{vr}$	$-2.00\times 10^{-4}$	-	$-2.00\times 10^{-4}$
$a_{ru}$	$-3.00\times 10^{-3}$	-	$-3.00\times 10^{-3}$
$a_{rv}$	$-1.00\times 10^{-2}$	-	$-1.00\times 10^{-2}$
$a_{rr}$	$-1.18\times 10^{-2}$	$-0.37\times 10^{-2}$	$-7.75\times 10^{-3}$
$b_{uu}$	$2.11\times 10^{-3}$	$6.12\times 10^{-3}$	$3.62\times 10^{-3}$
$b_{uv}$	$2.06\times 10^{-3}$	-	$2.06\times 10^{-3}$
$b_{vr}$	$-1.28\times 10^{-5}$	$4.49\times 10^{-5}$	$1.61\times 10^{-5}$
$b_{ru}$	$3.00\times 10^{-5}$	-	$3.00\times 10^{-5}$
$b_{rv}$	$1.15\times 10^{-5}$	-	$1.15\times 10^{-3}$
$b_{rr}$	$8.00\times 10^{-5}$	-	$8.00\times 10^{-3}$

**Table 3 sensors-20-03533-t003:** Comparison of mean deviation from given values describe in (Equation 34).

Manouvre	Signal	Robust Control Sim Lake	Linear Matrix Inequalities Sim Lake
1	$\Delta_{u}[\%]$	0.0280	0.0167
2	$\Delta_{u}[\%]$	0.0385	0.0282
	$\Delta_{v}[\%]$	0.0249	0.0306
3	$\Delta_{u}[\%]$	0.0086	0.0088
	$\Delta_{v}[\%]$	0.0137	0.0079
	$\Delta_{r}[\%]$	0.0627	0.0760

**Table 4 sensors-20-03533-t004:** Comparison of mean deviation values after a time when received values are diffrent from given values describe in (Equation 35).

Manouvre	Signal	Robust Control Sim Lake	Linear Matrix Inequalities Sim Lake
1	$\Delta_{u}\left[\%\right]$	2.0897	3.6033
2	$\Delta_{u}\left[\%\right]$	2.3078	3.8103
	$\Delta_{v}\left[\%\right]$	4.0525	3.9736
3	$\Delta_{u}\left[\%\right]$	1.0119	2.2886
	$\Delta_{v}\left[\%\right]$	2.7124	2.0503
	$\Delta_{r}\left[\%\right]$	8.6222	8.8481

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
