# Peer review of "Effectiveness of Multidimensional Controllers Designated to Steering of the Motions of Ship at Low Speed"

_sensors, 2020, doi:10.3390/s20123533_

Round 1

Reviewer 1 Report

The manuscript aims to solve the problem of ship motion control at low speed, a novel multidimensional controller is proposed for an autonomous floating model “BLUE LADY” with uncertainties. The topic is interesting, valuable and interesting. First, the nonlinear model of the ship was built for controllers synthesis using the Matlab package. Then, the LMI controller is developed for the augmented model of the “BLUE LADY”. Finally, closed-loop performance simulation and comparison simulation are conducted to demonstrate the effectiveness of the proposed algorithm. Though, there are some problems in the current version, which should be checked and revised by authors.

  1. It is recommended that the abstract should be reorganized, since the content of the manuscript is not summarized clearly. The contribution of authors should be illustrated more detailly.
  2. Because the Journal name is Sensors, I think the sensors used in your experiment and their functions should be stated.
  3. Though there are lots of LMI method related reference cited, reference about the model identification are still lack in the introduction. For the purpose to complete the literature reviews, the following reference can be cited.

[1] Zhang Guoqing, Zhang Xianku, Pang Hongshuai. Multi-innovation auto-constructed least squares identification for 4 DOF ship manoeuvring mode modeling with full-scale trial data. ISA Transactions, 2015, 58: 186-195.

[2] W. Ramire, Z. Leong, H. Nguyen et al. Non-parametric dynamic system identification of ships using multi-output Gaussian Processes. Ocean Engineering, vol.166,no.10,pp.26-36,OCT,2018,DOI. 10.1016/j.oceaneng.2018.07.056.

[3] M. Zhu, A. Hahn and Y. Wen. Identification-based controller design using cloud model for course-keeping of ships in waves. Engineering Applications of Artificial Intelligence, vol.75,no.10,pp.22-35,OCT,2018,DOI. 10.1016/j.engappai.2018.07.011.

[4] J. Park, H. Sung and F. Ahmad et al.  A Numerical Identification of Excitation Force and Nonlinear Restoring Characteristics of Ship Roll Motion. Journal of Marine Science and Technology-Taiwan, vol.25,no.4,pp.475-481, AUG,2017,DOI. 10.6119/JMST-017-0418-1.

[5]Zhang Zhiheng, Zhang Xianku, Zhang Guoqing. ANFIS-based course-keeping control for ships using nonlinear feedback technique. Journal of Marine Science and Technology, 2019, 24: 1326-1333.

  1. As the author illustrated in Fig.1, the “BLUE LADY” is equipped with 5 actuators, but the actual control input for each actuator is unknown. It is recommended that the thrust allocation strategy should be embedded in the proposed control algorithm.
  2. In order to facilitate the reader’s understanding, it is recommended that more detailed explanations should be illustrated for the experiment results.
  3. There are some typos and grammatical errors in the manuscript, the authors should check it thoroughly before submitting the manuscript. Such as:
  4. Spelling mistake in the abstract, "byt".
  5. In Table 2, why using "÷" to represent range?
  6. In line 158,175,248,249, “nxn”, “3x3”, “21x14”, “3x6”, use multiplication sign rather than ‘x’. Check other multiplication sign.
  7. Line 158. The i in Fi should be subscripted, the dots should have the number of three.
  8. About reference [42], it is better to cite it as a research about heading stabilization. Although this backstepping process is able to have connection with CGSA algorithm[1], which is an algorithm simplified from H-inf robust control, as far as in [42] itself, the focus is not on robust control. It is about a SISO system for heading control.

[1]Xian-ku Zhang, Xu Han, Wei Guan, Guo-qing Zhang. Improvement of integrator backstepping control for ships with concise robust control and nonlinear decoration. Ocean Engineering, 2019, 189(1). https://doi.org/10.1016/j.oceaneng.2019.106349.

  1. Several symbols in Figure 6 do not have corresponding explanations. The “hats” are not the same as in line 91.
  2. Figure 7 is not clear enough. Use higher resolution graph.

Based on above consideration, the manuscript is suggested minor revised.

Author Response

Dear Reviewer, thank you for your revision and comments, this is very important feedback for us.

  1. It is recommended that the abstract should be reorganized, since the content of the manuscript is not summarized clearly. The contribution of authors should be illustrated more detailly.

        Abstract has been reorganized and summarized more clearly. Both authors are part of a research team and all work was common.

2. Because the Journal name is Sensors, I think the sensors used in your experiment and their functions should be stated.

        Information about used sensors has been added in text lines 102 to 108 and 252 to 257.

3.Though there are lots of LMI method related reference cited, reference about the model identification are still lack in the introduction. For the purpose to complete the literature reviews, the following reference can be cited.

      This was added to bibliography and changed in text lines 52-63

4. As the author illustrated in Fig.1, the “BLUE LADY” is equipped with 5 actuators, but the actual control input for each actuator is unknown. It is recommended that the thrust allocation strategy should be embedded in the proposed control algorithm.

      This information was added in text lines 93-95.

5. In order to facilitate the reader’s understanding, it is recommended that more detailed explanations should be illustrated for the experiment results.

      Experiment result details and explanations have been added in text lines 252-257. 

6. There are some typos and grammatical errors in the manuscript, the authors should check it thoroughly before submitting the manuscript. Such as:

  • Spelling mistake in the abstract, "byt".
  • In Table 2, why using "÷" to represent range?
  • In line 158,175,248,249, “nxn”, “3x3”, “21x14”, “3x6”, use multiplication sign rather than ‘x’. Check other multiplication sign.
  • Line 158. The i in Fi should be subscripted, the dots should have the number of three.

        All above mistakes corrected.

10. About reference [42], it is better to cite it as a research about heading stabilization. Although this backstepping process is able to have connection with CGSA algorithm[1], which is an algorithm simplified from H-inf robust control, as far as in [42] itself, the focus is not on robust control. It is about a SISO system for heading control.

       This has been corrected in text line 74.

  1. Several symbols in Figure 6 do not have corresponding explanations. The “hats” are not the same as in line 91.
  2. Figure 7 is not clear enough. Use higher resolution graph.

       All above mistakes have been corrected.

Reviewer 2 Report

Comments are provided in the attached file.

Author Response

Dear Reviewer, thank you for your revision and comments, this is very important feedback for us.

2. Also, a number of words are misused. For instance, on line 24 the authors refer to “low” speed as “small” speed. Also, at several spots, a “state space controller” is called a “space state controller”. The authors need to review the manuscript carefully to avoid such misuse of words.

It has been corrected.

  1. The authors need to spell out all acronym, such as ARPA (on line 22), ROV (on line 28), DSP (on line 30), etc. at the instances of their first appearance within the text.

All acronyms have been explained.

2. The authors need to define all variables, such as (u, v, r) (on line n=88), (τx, τy, τr) ((in equation 1), and the propulsion system control signals on line 89, etc. at the instances of their first appearance within the text.

It has been corrected.

3. From the literature review presented in Section 1, the drawbacks of the existing ship-steering controller design methods are not clear. The authors need to clearly explain the main drawbacks of the existing methods that are being addressed in this paper.

It was changed in line 52-63

4. The relationship between u and T (in equation 1 and line 89) needs to be made clear

It was changed in line 114-116, and Allocation is not directly the topic of this paper and as such is described in detail in [22] in bibliography.

5. Table 2 should be introduced first before presenting Table 1.

It has been corrected.

6. What do the state variables in equation (2) represent? Do they represent any physical variables?

They represent model coefficients received during model identification process. They are not physical values.

7. The authors need to clearly explain the dynamical ship model described by equations (1)-(5) in Section 2.

It is of course an important aspect of the paper but it is not directly its topic and as such it is explained in detail in quoted papers: [14], [34], [36], [39].

8. The presentation of a robust controller in Section 3 and an LMI controller in Section 4 are extremely poor. Both sections need major revisions to present the main steps of such controller design and how they are implemented here.

It is the opinion of the authors that detailed explanation of robust and LMI controller design would expand the volume of this paper beyond reasonable limits. That is why several papers explaining this are quoted.

9. In contrast to common convention, the signal flow in Figure 4 seems to be from right to left, which should be corrected. Also, there is a discrepancy between equation (8) and Figure 4.

It has been corrected.

10. The captions of Figures 5 and 6 do not describe what is shown in those figures.

It has been corrected.

11. What does equation (15) mean? Also, what does equation (21) mean?

The stability of the whole closed-loop control system is ensured by execution of the modified Small Gain Theorem. For described system (given by matrix Ted with uncertainties one can couch so called Structured Singular Value - ${Ted}$ (see equation (15)). Without a loss in generality, when "muTed" is smaller the 1, the control system is internally stable (more details one can found in Skogestad and Postlethwaite (2003) with the proofs of the proper theorems).

The above information has been added in text lines 174-178.

12. In Section 5.3, the authors refer to Figures 6.9 to 6.16, which are not identifiable in the paper.

It has been corrected.

13. In section 5.3, the units should be written as m/s, deg/s, etc.

It has been corrected.

14. In terms of performance, how do the proposed controllers compare with some of the existing methods?

Authors of this paper are part of a research team working on multidimensional control algorithms for Blue Lady ship model. Below authors [1] & [2] have, in the past, been working on single dimensional control of this ship model. It is the goal of this paper to show the the next step of research performed on this ship model and compare two methods of multidimensional control. Future research will include implementing algorithms for autopilot performing maneuvers according to IMO MASS 4.

[1] Morawski, L., Vinh, N. C., Pomirski, J., & Rak, A. (2007). The ship control system for trajectory tracking experiments with physical model of tanker. Polish Maritime Research.

[2] Morawski L., Pomirski J., Rak A., “Non-linear Control of Course-Unstable Ship: Experiments with Physical Tanker Model”, Proceedings of 10th IMAM Congress, Rethymnon, Crete, 2002.

Reviewer 3 Report

In this paper, the authors described two MIMO controllers applied to steer a training ship. Both multidimensional robust controller and LMI controller ensure the stability of the closed-loop nonlinear control system and also result to similar good performance in either simulation or experimental test. This paper is well organized and effective with quality.

Minor comments:

  1. For completeness, the proof of system robust stability should be included.
  2. The comparison of the proposed advantages with previous approaches is missing.

Author Response

Dear Reviewer, thank you for your revision and comments, this is very important feedback for us.

  1. For completeness, the proof of system robust stability should be included.

The stability of the whole closed-loop control system is ensured by execution of the modified Small Gain Theorem. For described system (given by matrix Ted with uncertainties one can couch so called Structured Singular Value - ${Ted}$ (see equation (15)). Without a loss in generality, when "muTed" is smaller the 1, the control system is internally stable (more details one can found in Skogestad and Postlethwaite (2003) with the proofs of the proper theorems). 

Above information was added in line 174 - 178.

2. The comparison of the proposed advantages with previous approaches is missing.

Authors of this paper are part of a research team working on multidimensional control algorithms for Blue Lady ship model. Below authors [1] & [2] have, in the past, been working on single dimensional control of this ship model. It is the goal of this paper to show the the next step of research performed on this ship model and compare two methods of multidimensional control.

[1] Morawski, L., Vinh, N. C., Pomirski, J., & Rak, A. (2007). The ship control system for trajectory tracking experiments with physical model of tanker. Polish Maritime Research.

[2] Morawski L., Pomirski J., Rak A., “Non-linear Control of Course-Unstable Ship: Experiments with Physical Tanker Model”, Proceedings of 10th IMAM Congress, Rethymnon, Crete, 2002.

Round 2

Reviewer 2 Report

Although the revised version addresses some of the concerns, it still has some shortcomings, as described below.

a) The following paragraph is quoted from the authors’ response:

3. From the literature review presented in Section 1, the drawbacks of the existing ship-steering controller design methods are not clear. The authors need to clearly explain the main drawbacks of the existing methods that are being addressed in this paper.

It was changed in line 52-63

Comment:

Although lines 52-63 of the revised manuscript cite only a few of many existing articles related ship steering control problem, it does not mention their drawbacks and doesn’t discuss how those drawbacks are addressed in this paper. 

b) The following paragraph is quoted from the authors’ response:

14. In terms of performance, how do the proposed controllers compare with some of the existing methods?

Authors of this paper are part of a research team working on multidimensional control algorithms for Blue Lady ship model. Below authors [1] & [2] have, in the past, been working on single dimensional control of this ship model. It is the goal of this paper to show the the next step of research performed on this ship model and compare two methods of multidimensional control. Future research will include implementing algorithms for autopilot performing maneuvers according to IMO MASS 4.

[1] Morawski, L., Vinh, N. C., Pomirski, J., & Rak, A. (2007). The ship control system for trajectory tracking experiments with physical model of tanker. Polish Maritime Research.

[2] Morawski L., Pomirski J., Rak A., “Non-linear Control of Course-Unstable Ship: Experiments with Physical Tanker Model”, Proceedings of 10th IMAM Congress, Rethymnon, Crete, 2002.

Comment:

This doesn’t answer what was asked above. Since the authors are introducing a couple of new ship steering controller design methods here, as a reader I would expect to see some results that indicate how does the performance of the proposed methods compare with some of the existing methods.

Author Response

Dear reviewer:

3. From the literature review presented in Section 1, the drawbacks of the existing ship-steering controller design methods are not clear. The authors need to clearly explain the main drawbacks of the existing methods that are being addressed in this paper.

It was changed in line 52-63

Comment:

Although lines 52-63 of the revised manuscript cite only a few of many existing articles related ship steering control problem, it does not mention their drawbacks and doesn’t discuss how those drawbacks are addressed in this paper. 

Comment author:

Controller synthesis methods, related to marine industry, presented in majority of publications are purely theoretical or based on computer simulations. The very few publications dealing with actual ships or ship models focus on single dimensional control, trajectory control responsible for moving the ship over a set path. Publications from Rolls Royce [40] and Konsberg [18] describe much more complex multidimensional control in MASS 4 standard but being published by industrial companies do not contain information about used control methods. In comparison this paper describes multidimensional control of a ship model during actuall sea trails with a detailed description of methods used.

It was changed in line 77-84

14. In terms of performance, how do the proposed controllers compare with some of the existing methods?

Authors of this paper are part of a research team working on multidimensional control algorithms for Blue Lady ship model. Below authors [1] & [2] have, in the past, been working on single dimensional control of this ship model. It is the goal of this paper to show the the next step of research performed on this ship model and compare two methods of multidimensional control. Future research will include implementing algorithms for autopilot performing maneuvers according to IMO MASS 4.

[1] Morawski, L., Vinh, N. C., Pomirski, J., & Rak, A. (2007). The ship control system for trajectory tracking experiments with physical model of tanker. Polish Maritime Research.

[2] Morawski L., Pomirski J., Rak A., “Non-linear Control of Course-Unstable Ship: Experiments with Physical Tanker Model”, Proceedings of 10th IMAM Congress, Rethymnon, Crete, 2002.

Comment:

This doesn’t answer what was asked above. Since the authors are introducing a couple of new ship steering controller design methods here, as a reader I would expect to see some results that indicate how does the performance of the proposed methods compare with some of the existing methods.

Comment author:

The two proposed multidimensional control methods can be used for autopilot operation and not just trackpilot like the previous single dimensional methods tested on Blue Lady ship model. Both Robust and LMI controllers are resistant to wind interference as well as GPS mode fluctuations in contrast to prior reasearch detaied in [1] and [2] where both of these were serious issues. Controller synthesis progress is also seen in controller matrix size, Robust controller being the earlier development has matrix size 24x21 while the later LMI controller matrix is of 6x3 size. Robust and LMI controller synthesis methods are considered classical, more modern methods as described in [51] have not been tested on Blue Lady ship model yet. 
